# The Endocannabinoid System of Animals

**DOI:** 10.3390/ani9090686

**Published:** 2019-09-16

**Authors:** Robert J. Silver

**Affiliations:** Chief Medical Officer, RxVitamins, Niwot, CO 80544-0590, USA; rsilver@drsilverdvm.com

**Keywords:** endocannabinoid system, Anandamide, 2-AG, cannabis, cannabinoid receptor 1, cannabinoid receptor 2, G-coupled protein receptor, PPARS a, b, Ht1a, TRPV1, GPR55, cannabidiol, CBD, THC, CBG, CBC, tetrahydrocannabinol

## Abstract

**Simple Summary:**

Our understanding of the Endocannabinoid System of animals, and its ubiquitous presence in nearly all members of *Animalia*, has opened the door to novel approaches targeting pain management, cancer therapeutics, modulation of neurologic disorders, stress reduction, anxiety management, and inflammatory diseases. Both endogenous and exogenous endocannabinoid-related molecules are able to function as direct ligands or, otherwise, influence the EndoCannabinoid System (ECS). This review article introduces the reader to the ECS in animals, and documents its potential as a source for emerging therapeutics.

**Abstract:**

The endocannabinoid system has been found to be pervasive in mammalian species. It has also been described in invertebrate species as primitive as the Hydra. Insects, apparently, are devoid of this, otherwise, ubiquitous system that provides homeostatic balance to the nervous and immune systems, as well as many other organ systems. The endocannabinoid system (ECS) has been defined to consist of three parts, which include (1) endogenous ligands, (2) G-protein coupled receptors (GPCRs), and (3) enzymes to degrade and recycle the ligands. Two endogenous molecules have been identified as ligands in the ECS to date. The endocannabinoids are anandamide (arachidonoyl ethanolamide) and 2-AG (2-arachidonoyl glycerol). Two G-coupled protein receptors (GPCR) have been described as part of this system, with other putative GPC being considered. Coincidentally, the phytochemicals produced in large quantities by the *Cannabis sativa* L plant, and in lesser amounts by other plants, can interact with this system as ligands. These plant-based cannabinoids are termed phytocannabinoids. The precise determination of the distribution of cannabinoid receptors in animal species is an ongoing project, with the canine cannabinoid receptor distribution currently receiving the most interest in non-human animals.

## 1. Introduction

Common to nearly all animals except the Phyla *Protozoa* and *Insecta*, the endocannabinoid system arose in the phylogeny concurrently with the development of the nervous system as multicellular animals developed increasing complexity. This system was unknown to scientists until the mid-1990s, but research into this fascinating and clinically useful system is advancing rapidly, especially with the use of state-of-the-art LC-MS/MS analyzers as well as immunohistochemical and polymerase chain reaction (PCR) analytic technologies.

For more than 70 years, scientists have been hobbled by the legal and regulatory prohibitions related to research into cannabis and its associated molecules. Phytocannabinoids, from *Cannabis sativa* L.primarily, and also found in a few other plant sources, are exogenous plant-based ligands that interact with cannabinoid receptors.

## 2. The Endocannabinoid System

Found in nearly all animals, from mammals to the more primitive phyla such as Cnidaria, the early emergence of the ECS in the evolution of the Phyla, indicates its biological importance. All animals, including vertebrates (mammals, birds, reptiles, and fish) and invertebrates (sea urchins, leeches, mussels, nematodes, and others) have been found to have endocannabinoid systems.

The Hydra (*H. vulgaris*), a cnidarian in the class Hydrozoa, is one of the first animals with a neural network. The major function of the ECS in this primitive organism was determined by De Petrocellis in 1999, to control its feeding response [1]. Since the evidence indicates that all veterinary species have an ECS, an understanding in these species of the role of the endocannabinoid system will be vital for developing clinical applications, both for the endogenous cannabinoids and associated molecules, and the plant molecules derived primarily from *Cannabis sativa* L., such as the phytocannabinoids, terpenes, and flavonoids.

The ECS was discovered secondary to the discovery of the structure of the psychotropic phytocannabinoid, -Δ-9-tetrahydrocannabinol (THC). Cannabinoid receptor 1 was found during the search for the biological target(s) for THC [2]. THC is the only psychotropic cannabinoid found in *Cannabis sativa* L and is responsible for some of this plant’s biomedical activity. The non-psychotropic cannabinoids such as cannabidiol (CBD), Cannabigerol (CBG), Cannabichromene (CBC), other minor cannabinoids, terpenes, and flavonoids have been found to have comparable biomedical activity to that of THC without its side-effect of intoxication.

Studies at the National Institute on Drug Abuse in Bethesda, Maryland cloned the G-coupled protein receptor (GPCR) in 1990, which is the target for endogenous cannabinoid ligands, and named it, “Cannabinoid Receptor 1 (CB_1_ or CBR_1_). This receptor belongs to the Class A rhodopsin-like family of GPCRs [3]. A few years later, the second GPCR: “Cannabinoid Receptor 2” (CB_2_ or CBR_2_) was cloned [4]. The CB_1_ and the CB_2_ receptors participate in numerous essential biological processes [5]. Some of these are: Neuronal plasticity [6], pain [7], anxiety [8], inflammation [9], neuro-inflammation [10], immune function [11], metabolic regulation [12], and bone growth [13].

Following the discovery of the membrane receptors that accept plant-based ligands, researchers quickly identified the endogenous ligands that bind to the cannabinoid receptors (CBR). These endogenous ligands are the endocannabinoids (eCB), arachidonoyl ethanolamide (AEA), a long-chain fatty acid amide, which was named “anandamide” by Mechoulam in 1992. “Ananda” in Sanskrit means “bliss.” This endocannabinoid engenders feelings of well-being, and, since its discovery, is now considered to be the “bliss molecule” responsible for the “runner’s high” that many athletes describe. The ester of this fatty acid amide, 2-arachidonoyl glycerol (2-AG), was discovered in 1995 [14,15,16]. Both of these compounds can bind to either of the endocannabinoid receptors (CB_1_ and CB_2_). THC is the only phytocannabinoid that binds orthosterically to these endocannabinoid receptors.

eCBs are produced ad hoc by enzymes located in the cell membrane stimulated by the intracellular elevation of calcium ions secondary to neuronal depolarization. The constitutive level of endocannabinoids is termed the “Endocannabinoid Tone”, and that level varies based on the specific tissue in which they are found. These levels are dependent upon their rate of production minus the rate of their enzymatic degradation. Fatty acid amide hydrolase (FAAH) and monoacyl-glycerol lipase (MAGL) are the specific enzymes involved in modulating endocannabinoid tone. AEA and 2-AG, which are the two endogenous endocannabinoids, are enzymatically metabolized by FAAH and MAGL, respectively. FAAH can also metabolize 2-AG, although to a lesser extent.

The ECS is divided into the three categories shown below.
Endogenous ligandsMembrane receptorsDeactivating enzymes

A transport protein mechanism carries eCBs retrograde from post-synaptic cell membranes to bind to the cannabinoid receptors (CBR) found on the pre-synaptic membrane. This transport mechanism then returns the eCBs to the post-synaptic membrane where they are degraded by the enzymes FAAH or MAGL, for anandamide and 2-AG..

Endocannabinoids act on the presynaptic endocannabinoid receptors following their release from the post-synaptic neuronal membrane. The eCBs modulate neurotransmitter release by inhibiting the influx of intracellular calcium, which, in turn, inhibits the release of neurotransmitters. eCBs are rapidly re-absorbed and are then catabolized very quickly, which results in a very short half-life. eCBs are produced ad hoc after being stimulated by trauma, or by large amounts of nerve cell depolarization. eCB production occurs locally in the cell membrane, and primarily influences the contiguous tissue. The eCBs have fairly brief activity. They are subject to rapid re-uptake by the cell and then are degraded by enzymes that are part of the endocannabinoid system. Another endocannabinoid transporter mechanism carries the eCBs into the cell where it accumulates and, subsequently, leads to its enzymatic degradation. Anandamide, for instance, is degraded by fatty acid hydrolase (FAAH) and converted to arachidonic acid and ethanolamine [17,18].

To date, there are only two types of signaling that have been recognized in the endocannabinoid system: phasic signals and tonic signals. Tonic signaling, which is also known as “basal signaling,” is responsible for the endocannabinoid tone. When eCB levels change over time, this is responsible for “phasic signaling.

Polyunsaturated fatty acids are known to directly input into eCB signaling pathways, which makes intake of omega three fatty acids necessary in the regulation of ECS tone [19]. The ECS is intimately involved in the regulation of most aspects of animal physiology. The CB_1_ cannabinoid receptor has been found to be the most common GPCR in the human brain, and in many other organs. To date, these anatomical locations for the CB_1_ receptor includes: heart, blood vessels, liver, lungs, digestive system, fat, and sperm cells [20].

The CB_1_ receptor, as a member of the family of G-Coupled Protein Receptors (GPCR), is categorized as a Class A rhodopsin-like receptor. Anatomically, the central nervous system is its primary locus, with high concentrations in the cortex, hippocampus, outflow of the basal ganglia, and cerebellum. Intraspecies and interspecies differences are common in terms of the anatomical sites and density of cannabinoid receptors. These receptors, in humans, are not located to any extent in the brain stem or medulla oblongata, which control vital autonomic functions such as respiration and heart rate. It is the absence of any significant amount of these receptors in areas that control these vital functions that is responsible for the safe profile of cannabinoids for humans [21].

One interspecies variation in the anatomical location of the CB_1_ receptors is found in dogs. As compared to humans, studies have determined the number of CB_1_ receptors in hind brain structures in the dog to far exceed those found in the human animal. The US government conducted studies that determined that dogs have large numbers of cannabinoid receptors in the cerebellum, brain stem, and medulla oblongata [22]. “Static ataxia,” which is a unique neurological reaction to THC in the dog, is explained by this high concentration of CB receptors in the cerebellum. Static ataxia was first described in 1899 by Dixon in his pharmacologic study of Indian Hemp (High THC cannabis) in a variety of species, including human [23]. Other locations for Cannabinoid receptor 1 include the peripheral nervous system, as well as cardiovascular, immune, gastrointestinal, and reproductive tissues. Cannabinoid receptor 2 has been found mainly in cells of the immune system and the spleen and tonsils [24]. The CB_1_ and CB_2_ receptors are structurally quite similar, despite their different anatomical locations in the central nervous and immune systems, respectively.

CB_2_ receptors in the immune system can modulate the release of cytokines. Inhibition of adenyl cyclase results from the activation of lymphocyte CB_2_ receptors by cannabinoids. This, in turn, can reduce the cellular and humoral responses to an immune challenge [25]. CB_1_ and CB_2_ receptors will decrease adenyl cyclase activity and down-regulate the cAMP pathway. Other results of activation of lymphocytes include mitogen-activated protein kinase (MAPK) cascades, ion channel modulation, and modification of intracellular calcium levels, which leads to a neurotransmitter release [26,27,28,29]. In addition to intracellular calcium levels, potassium channel activation is also a signaling mechanism for the CB_2_ receptor [30,31].

Fatty acid-binding proteins (FABP) are needed to transport endocannabinoids into the cell where they bind to cannabinoid receptors on the outer mitochondrial membrane. These mitochondrial receptors result in two mechanisms: (1) They recruit nuclear transcription factors to alter gene expression [32], and (2) they regulate the metabolism of neurons [33]. The cannabinoid receptor interacting protein, cannabinoid interacting protein (CRIP1a) inhibits tonic (constitutive) eCB signaling [34]. These mitochondrial CB_1_ receptors alter the energy metabolism of the cell by reducing the activity in the electron transport chain and through inhibition of soluble adenyl cyclase. Mitochondrial receptors release ceramide, which plays a role to create Endoplasmic reticulum (ER) stress in cancer cells and leads to autophagy and potentially apoptosis and cell death [35].

## 3. Veterinary ECS: Our Current State of Knowledge

Compared to the studies and information available about both the medical and the health benefits derived from cannabinoids in the human animal, there is still a paucity of information regarding the same benefits in animals, except for the laboratory animal species in which experimental studies have been performed. It is known, though, that the endocannabinoid system, which is universal to all animal species except insects, possesses essentially the same benefits regardless of the species under review.

Differences in the protein sequences of the CB_2_ receptor have been identified in the human, rat, and canine receptors. This is true despite the fact that the CB_1_ receptor structure remains similar among all mammalian species. Endogenous ligand binding affinities for the canine CB_2_ receptor have been measured to be about 30 times less than human and rat CB_2_ receptors. The function of Cannabinoid Receptor 2 is dependent upon its level of expression on cell membranes, and the type of signaling pathways involved [36].

### 3.1. Anatomical Localization of Cannabinoid Receptors in the Dog

#### 3.1.1. Cannabinoid Receptor 1

The anatomical localization of the CB_1_ receptor in the normal canine nervous system has been determined through the use of immunohistochemical analysis. Currently, studies are underway using the more accurate polymerase chain reaction (PCR) technology, but that data is, as yet, unpublished. Nervous systems from healthy dogs at 4 months, 6 months, and 10-year-old dogs were evaluated post-mortem. Neutrophils of the cerebral cortex, cornu ammonis (CA), dentate gyrus of the hippocampus, midbrain, cerebellum, medulla oblongata, and gray matter of the spinal cord were found to have strong immunoreactivity. Dense CB_1_ expression was found in the fibers of the globus pallidus and substantia nigra, based upon dense CB_1_ immunoreactivity. These immuno-reactive locations were surrounded by neurons with no immunoreactivity.

A consistent finding of positive immunoreactivity in astrocytes was recorded in all of the examined regions. In the peripheral nervous system, CB_1_ staining was localized in the neurons and in the satellite cells of the myelinating Schwann cells and dorsal root ganglia.

Comparing the younger dog nervous system to the older dog nervous system, lower CB_1_ expression was found in the brain tissue. This was less than the expression of receptor immunohistochemistry found in human fetal and neonatal brain tissue. Reduced receptor expression has been measured in aged rats, localized to the cerebellum, cerebral cortex, and basal ganglia and present, but less prevalent in the hippocampus. The older dog in this study was also found, like the aged rats, to have reduced measurements of CB_1_ receptor expression compared to the younger dogs’ nervous systems examined [37]. CB_1_ receptors in salivary glands [38], hair follicles [39], skin, and the hippocampus in dogs have been previously described [40].

When a 30-day old canine embryo was examined using immunohistochemistry to localize its CB_1_ receptors, immunoreactivity was identified mainly in epithelial tissues and included most structures of the central and peripheral nervous system, inner ear, olfactory epithelium and related structures, eyes, and thyroid gland [41].

Summary of CB_1_ Receptor Studies in the Dog: Anatomical and Cellular Localization [39]
Basal and supra-basal layer cell cytoplasmThe inner epithelial root sheaths and arrector pili muscles of the hair folliclesSebaceous gland sebocytes that are undifferentiated and located at the gland peripheryFibroblastMast cellsCB1 receptors were found to be up-regulated in canine patients with atopic dermatitis

#### 3.1.2. Cannabinoid Receptor 2

A homogenous distribution of both CB_1_ and CB_2_ receptors in clinically normal dogs is found throughout all layers of the epidermis. In the human epidermis, the CB_1_ receptor was localized to the epidermal spinosum and granulosum layers, and the CB_2_ receptor has been identified in the basal keratocytes. CB_1_ and CB_2_ receptors are present both in healthy dog epidermis and in dogs with atopic dermatitis. There is a basic anatomical difference between human and canine epidermal architecture, with canine epidermis containing two to three nucleated layers of cells, whereas human epidermis contains six to seven nucleated layers of cells. Dogs diagnosed with atopic dermatitis have a hyperplastic epidermis. Suprabasal keratinocytes possess strong immunoreactivity for both the CB_1_ and CB_2_ receptors, whereas there was weak CB_1_ but strong CB_2_ immunoreactivity in basal keratinocytes. This is a strong indication that these receptors are up-regulated during epidermal inflammation. Agonists to both CB_1_ and CB_2_ receptors have been found to reduce mast cell degranulation, which is an important step in the development of hypersensitivity reactions.

Canine CB_2_ Receptor Epidermal Localization [39]
Basal and supra-basal layer cellular cytoplasmHair follicles◦Basal and supra-basal cells◦Outer and inner epithelial root sheathsArrector pili musclesSweat glands◦Secretory and ductal cellsSebaceous glandsMast cells, fibroblasts, and endothelial cellsLymph nodes◦Strong B-cell zone immunoreactivity◦Primarily in the terminal centers of secondary folliclesUpregulated in atopic dermatitis

### 3.2. Invertebrate Endocannabinoid Systems

Identification of the cannabinoid receptors in non-mammalian species have found their presence in birds, reptiles, and fish. In a study of seven representative species of invertebrates, McPartland 2006, evaluated seven species of invertebrates by a tritiated ligand binding assay and identified cannabinoid receptors in the following species.
*Ciona intestinalis* (Deuterostomia)*Lumbricus terrestris* (Lophotrochozoa)*Peripatoides novae-zealandiae* (Onychophora)*Jasus edwardi* (Crustacea),*Pangrellus redivivus* (Nematoda) [the beer mat nematode]*Actinothoe albocincta* (Cnidaria) [white striped anemone]*Tethya aurantium* (Porifera) [Orange Puffball sponge] [42].

Two species that did not show evidence of cannabinoid binding were the sea anemone (*A. albocincta*) and sponge (*T. aurantium*). CB_1_ receptors were detected but CB_2_ receptors were not detected. The earthworm (*L. terrestris*), velvet worm (*P. novae-zealandiae*), and mat nematode (*P. redivivus*) were compared to a standard CB_1_ ortholog in rat cerebellar tissue and were found to have high affinity binding interactions at concentrations typically found with CB1 receptors.

McPartland hypothesized that cannabinoid receptors evolved in the last common ancestor of the bilaterians, but had secondary loss in insects and other clades. Cannabinoid receptors have been identified in sea urchins, leeches, earthworms, hydra, lobster (*H. americanus* and *J. edwardi*), and the beer mat nematode (*P. redivivus*), but not the nematode (*C. elegans*). No cannabinoid binding has been observed in sponges (Porifera).

Insects (*Apis mellifera* [western honey bee]), *Drosophila melanogaster* [common fruit fly], *Gerris marginatus* [water strider], *Spodoptera frugiperda* [fall armyworm moth larva], and *Zophobas atratus* [darkling beetle]) do not contain cannabinoid receptors. No other mammalian neuroreceptor has been found to be lacking in insects. This is the only case in comparative neurobiology that a mammalian neuroreceptor is absent in insects (*Ecdysozoa*). One hypothesis for this absence of cannabinoid receptors in insects is their lack of endocannabinoid ligands. Insects manufacture minimal to no arachidonic acid, which is a precursor in the body’s manufacturing of its endocannabinoids [43].

### 3.3. The Endocannabinoid System and Disease

There is a paucity of studies in our veterinary species regarding the relationship between the endocannabinoid system and specific diseases. The existing published studies have focused on the human animal or have utilized laboratory animal experimental models. There is a significant need for clinical studies in veterinary species to provide evidence-based applications for phytocannabinoid and endocannabinoid molecules [21].

#### 3.3.1. Modulation of Anxiety and Stress

It has been theorized that certain chronic conditions, such as Post Traumatic Stress Disorder (PTSD), migraine, fibromyalgia, and inflammatory bowel disease are the result of deficiencies in tonic eCB signaling. Circulating levels of eCBs are markedly decreased in these disorders. This diminished amount of eCBs has been correlated to anxiety-like behaviors. It is known, for instance, that chronic environmental stress will down-regulate CB_1_ receptors as well as reducing levels of AEA (anandamide) at the same time as increasing levels of 2-AG [44].

One study examined people who were administered a “stressful public speaking test” (SPST). These subjects were administered 300 mgs of purified CBD isolate to treat their anxiety, which was determined to have effects that were comparable to the anxiolytic pharmaceutical benzodiazepam (Valium™) [45]. CBD can counteract the undesirable effects of THC when administered concurrently.

#### 3.3.2. Modulation of Inflammatory Conditions

Inflammation is the causal root of many diseases and health conditions. The immune system is, in part, regulated by the immunomodulatory effect of the endocannabinoid system as determined from in vivo and in vitro studies. Experimental models for a variety of autoimmune diseases have determined that cannabinoids are key players in multiple sclerosis, rheumatoid arthritis, colitis hepatitis, and psoriasis.

Cells of the immune system have been found to have the cannabinoid receptor CB_2_ present on their cell membrane. These CB2 receptors function to modulate a healthy response to inflammation by up-regulating several anti-inflammatory pathways, including the inhibition of T-cell pro-inflammatory activity.

T-cell anti-inflammatory mechanisms affected by CBR agonists:Apoptosis of T-cellsSuppression of pro-inflammatory cytokines and chemokinesInhibition of T-effector cell proliferationPromote the proliferation of T-regulatory cells.

The cannabis plant contains many molecules that reduce inflammation. The three major groups of molecules found in the cannabis plant are Phytocannabinoids, terpenes, and flavonoids, which all have strong anti-inflammatory properties. Phytocannabinoids activate CD95, which then induces both Bcl-2 and caspase cascades that lead to the apoptosis of immune cells. Phytocannabinoids can increase the production of IL-10, which has anti-inflammatory properties at the same time as they can reduce the manufacture of TNF-α and other pro-inflammatory cytokines [46,47].

Pharmaceutical NSAIDs use the endocannabinoid system in the creation of their anti-inflammatory effect. Acetaminophen is metabolized in the liver, and its by-product, *N*-arachidonoylphenolamine, functions both as a cannabinoid receptor agonist and as an eCB re-uptake inhibitor [48]. Arachidonic acid can be converted to pro-inflammatory eicosanoids. However, arachidonic acid is also the major precursor in the production of the anti-inflammatory endocannabinoids, anandamide, and 2-AG. Normally, COX enzymes speed up the degradation of anandamide, which contributes to pain and inflammation. Non-steroidal anti-inflammatory drugs are COX-2 inhibitors that work, in part, by reducing the enzymatic degradation of the endocannabinoid anandamide [49].

In addition to the anti-inflammatory activity of phytocannabinoids, the terpenes and terpenoids found in the cannabis plant also have potent anti-inflammatory activity. Their anti-inflammatory effect is caused by the binding of certain terpenes to prostaglandin receptors, PGE1 and/or PGE2.

Terpenes most responsible for anti-inflammatory activity are:α-pineneβ-myrceneβ-caryophyllenelimonene

Of the terpenes, both β-caryophyllene and limonene have been found to bind to cannabinoid receptors, which reduce inflammation in a CB_2_-receptor dependent fashion [50]. These two terpenes are partial agonists of the CB receptors, and can compete with THC for CB binding sites. Thus, they are both known to antidote THC excesses by replacing the THC bound to the CB receptor with one of these two terpenes.

Historically, lemon products and spicy products with black pepper and cloves have been used, due to their limonene and beta caryophyllene content, respectively, for excesses of THC, panic attacks, and tachycardia. As competitive agonists for the CB_1_ receptor, they force THC to separate from the G-Protein receptor in the central nervous system, which reduces the influence of THC, the panic attack, and the increased heart rate.

#### 3.3.3. Modulation of Pain

Endocannabinoids modulate the neural conduction of pain signals by both reducing the nociceptive neural signal of pain, and by reducing inflammation by activating cannabinoid receptors.

CB_1_ receptors are present on the cell membrane of both nociceptive and non-nociceptive sensory neurons found in the trigeminal ganglion and dorsal root ganglia. Macrophages and mast cells have been found to contain CB_1_ receptors on their cell membrane, as well as epidermal keratinocytes. CB_1_ receptors modulate a neurotransmitter release in the brain and spinal cord.

Hematopoietic cells contain CB_2_ receptors. CB_2_ receptors in the brain, spinal cord, and dorsal root ganglion have been found. CB_2_ receptors up-regulate with peripheral nerve damage. Cannabinoid receptors regulate neuroimmune interactions and interfere with inflammatory hyperalgesia.

The endocannabinoids anandamide and 2-AG are produced in tissue that has been injured, and activate cannabinoid receptors to suppress the sensitization of the nerve to nociceptive signals and/or to suppress inflammation. Anandamide modulates pain by: 1) inhibiting nociceptive signals at the synapse by activating CB_1_ receptors, by 2) becoming transformed by COX-2 enzymes into prostamides (pain-relieving molecules), and by 3) reducing inflammation by activating CB_2_ and other receptors. 2-AG plays a role in the descending modulation of pain during acute stress. Both molecules are produced as the body’s first response to tissue injury [51].

#### 3.3.4. Metabolic Effects

Satiety, in part, is modulated through the hypothalamic pro-opiomelanocortin (POMC) neurons. POMC neurons are inhibited by CB1 receptor activation. The result is an increase in appetite, which explains the hyperphagia associated with the use of high THC cannabis. The lack of signals to the brain indicating satiety comes from blocking the release of the α–melanocyte-stimulating hormone (α-MSH), which is an appetite suppressant signaling molecule. Orexin-A induces hyperphagia by increasing the levels of 2-AG. α-MSH is inversely related to the levels of Orexin A [52].

Cannabis users who have hyperphagia, eat more calories due to this inhibitory effect of THC. Paradoxically, based on published studies, users of cannabis are slimmer than non-users. The prevalence of obesity is lower in regular cannabis users compared to non-users, who are also less likely to be obese. This is true when variables such as age, sex, and tobacco smoking status are adjusted for. Fasting insulin and insulin sensitivity are improved for chronic users of cannabis, as compared to non-users [53]. These are the metabolic effects of the endocannabinoid system, due, in part, to the presence of cannabinoid receptors on the mitochondrial membrane.

#### 3.3.5. Cancer

The anti-neoplastic effects of the endocannabinoid system include the modulation of cell differentiation, cellular proliferation, tissue invasion, and angiogenesis. Phytocannabinoids such as THC and CBD can benefit cancer patients by increasing appetite and reducing nausea. These anti-neoplastic qualities of the ECS make it an essential component of most cancer therapies. By modulating gene expression, cannabinoids can reduce the growth of certain neoplasms.

A study with lung cancer found that CBD up-regulates intracellular adhesion molecules (iCAM), which reduces the potential for metastasis [54].

CBD reduces the production of pro-angiogenic factors in gliomas in a dose-dependent fashion. THC when co-administered with CBD synergistically inhibited proliferation and caused cell cycle arrest in vitro of human glioblastoma cells [55].

CBD down-regulates the expression of ID-1, which is a metastatic factor for breast cancer cells and most carcinomas [56].

#### 3.3.6. The Role of Antioxidants and Neuroprotection

Cannabinoids act as antioxidants and neuroprotectants. The US National Institute of Health (NIH) holds a patent on these compounds for this purpose [57]. By definition, antioxidants neutralize reactive oxygen species (ROS). Cannabinoids inhibit the release of glutamate by inhibiting voltage-gated calcium channels. Glutamate causes the depolarization of neurons. Ischemia and traumatic brain events cause the release of glutamate. Glutamate can be toxic when its levels are high and excessive, and neuronal cell death can ensue as a result of excitotoxic stress.

Antioxidants are commonly neuroprotective. Toxic ROS are produced following ischemia, and antioxidants will reduce their damaging effects. Most of the phytocannabinoids, including THC and CBD, are antioxidants [58]. THC and CBD can prevent glutamate neurotoxicity. This neuroprotection effect is separate from their CB receptor binding activity. THC and CBD reduce ROS in vitro when compared to common antioxidants such as ascorbate and butylated hydroxytoluene (BHT). CBD protects nervous tissue against cerebral ischemic injury [59]. In animal models of Alzheimer’s Disease, CBD was able to reduce Alzheimer’s-related neuro-inflammation [60].

Terpenes have strong antioxidant properties and can also serve as neuroprotectants. In a mouse model of cerebral inflammation, β-myrcene, which is a common terpene, provided protection from oxidative stress and histological damage induced by ischemia-reperfusion.

Three Cannabis Terpenes with the highest free radical quenching ability include:β-caryophyllenelimoneneβ-myrcene [61,62]

Most of the phytocannabinoids and terpenes found in cannabis have been found to have potent antioxidant properties [47].

The ECS is involved in the development of many neurodegenerative conditions. Cannabinoids have been shown to have neuroprotective properties, possessing the ability to reduce neuroinflammation, and promote neurogenesis [63,64,65].

In Alzheimer’s Disease (AD), cannabinoids are able to clear the toxic beta amyloid (Aβ) plaques associated with this disease. AD has been found to be associated with a loss of the body’s natural production of eCBs. This deficiency condition has been termed a Clinical Endocannabinoid Deficiency Syndrome (CEDS) [60].

CBD has also been shown to reduce the expression of genes implicated in the phosphorylation of the tau protein, which leads to the formation of neurofibrillary tangles that contribute to the progression of the disease. Furthermore, cannabinoids enhance the clearance of Aβ from the brain as well as prevent the inflammatory cascade that is produced by the accumulation of these mis-folded proteins intracellularly [66].

#### 3.3.7. Cardiovascular Modulation by the ECS

Cannabinoids modulate blood pressure and heart rate, which increases or decreases, depending upon local conditions [67,68,69,70].

CBD has direct effects on arteries, causing vasodilation, and hypotension. The hypotensive effect is mild, but could be problematic in cases of uncompensated cardiac disease. The antioxidant and anti-inflammatory benefits of CBD reduces the hyperglycemic damage to blood vessels secondary to Type 2 diabetes mellitus. CBD benefits systemic inflammatory processes as well as diabetic angiopathies [71].

CBD demonstrated anti-arrhythmic effects following coronary artery occlusion in rats. This study found that these anti-arrhythmic effects were mediated through non-receptor pathways that did not involve the CB_1_ receptor [72].

CB_1_ antagonists were observed to increase systemic blood pressure as well as left ventricular contractility in spontaneous hypertensive rats. In normotensive rats, modulation of their blood pressure, cardiac contractility and vascular after load is secondary to the inhibitory effect of CBD on the fatty acid amide hydrolase (FAAH) enzyme. By reducing the degradation of anandamide, systemic levels become increased of this endogenous CB_1_ agonist. CB_1_ antagonists inhibit these effects. CB_1_ antagonists were able to lower blood pressure better in the hypertensive rats than the normotensive rats. This may be due to the upregulation of CB_1_ receptors in heart and aortic endothelium in hypertensive rats, but not the normotensive cohort [73].

#### 3.3.8. Modulation of Pulmonary Function

Inhaled and oral THC can create bronchodilation for up to two hours following administration. CB_1_ receptor activation inhibits cholinergic contraction, which allows it to inhibit bronchospasms. This may be why asthma sufferers find some comfort from cannabis use [74]. In mouse models of inflammatory lung disease, pulmonary function tests improved and reduced inflammation. CBD decreased pulmonary inflammation and improved pulmonary function. Improved pulmonary function was also observed in a model of COPD [75]. Another study found that THC reduced the production of allergen-induced mucus [76].

#### 3.3.9. Clinical Endocannabinoid Deficiency Syndrome

Russo has postulated that several chronic conditions may be due to deficiencies in eCB signaling. PTSD, migraine, Irritable Bowel Syndrome (IBS), PTSD, and fibromyalgia are of particular note in this Clinical Endocannabinoid Deficiency Syndrome (CEDS) [77,78]. Endocannabinoid genes that mutate are known to help create these deficiencies, which helps explain the genetic component of these diseases. Other causes that can affect the constitutive tone, or systemic level of eCBs, include certain pharmaceuticals and diseases that can deplete eCB levels or interfere with eCB production. Human patients with mutations in CNR1 and DAGLA genes show signs of CEDS [79]. Human IBS patients with mutations in the CNR1 gene, were found to also have altered rates of colonic transit [80]. Impaired fear extinction has been identified with PTSD, in patients who were homozygous for a CNR1 mutation [81].

These disorders all have in common markedly decreased systemic levels of eCBs. Circulating eCB deficiencies are also inversely correlated with anxiety-like behaviors. Chronic environmental stressors will down-regulate CB_1_ receptors and reduce levels of both AEA and 2-AG [82].

As of this publication, the CEDS has not been identified or defined in our veterinary species.

### 3.4. The Safety of Cannabinoids in Dogs

As a CBR agonist, the psychoactive effects of THC are undesirable in all veterinary species. Dogs in particular will suffer from “Static Ataxia” upon exposure to THC at doses >0.5 mg/kg IV [[83]]. As a result of this exquisite sensitivity of dogs, in the United States, those states with medical or adult use marijuana laws have seen an increase in Animal ER admissions for THC toxicosis [84]. A report of the summary of calls to the Pet Poison Hotline found a relatively good safety profile for CBD, and THC, when given in moderate amounts [85]. Oral tolerance to THC can be achieved in the dog following 7–10 days of a low, sub-psychotropic dose of 0.05–0.1 mg/kg twice daily by mouth. This will remove the adverse neurological effects of THC, described as “Static Ataxia” but will still be medicinally effective, despite sedative effects at higher doses of THC [21]. Concurrent use of CBD in equal or greater amounts than THC will assist in this tolerizing process. In humans, ECS stimulation that is excessive and prolonged can create memory deficits. Upon cessation of prolonged ECS stimulation, withdrawal symptoms will develop. Withdrawal symptoms in dogs have been documented following tolerization, but impaired memory has not yet been studied in veterinary species [86]

A study conducted at Colorado State University, published in 2018, found CBD, when present in a full-spectrum hemp oil extract, to be safe when given at a high dose of 20 mg/kg/day for six weeks to Beagles. This group of 30 dogs were found to have mild elevations in serum alkaline phosphatase in about 30% of the subjects, and the side-effect of diarrhea in all of the subjects. As CBD is metabolized via the P450 enzyme system, there is concern that the concurrent use of CBD with drugs that are also metabolized through that pathway may have their pharmacokinetics altered, which could alter their therapeutic value for a given patient [87].

As of this publication, no definitive objective study exists accurately and objectively, which defines the herb-drug interactions between CBD and selected pharmaceuticals in veterinary species.

## 4. Discussion

From the research presented in this paper, it is clear that the endocannabinoid system is not just present in nearly all animals, but plays an integral role in maintaining homeostasis for a number of organ systems. The endocannabinoid system modulates the nervous and immune systems and other organ systems through a complex system of receptors and chemical signaling molecules to relieve pain and inflammation, modulate metabolism and neurologic function, promote healthy digestive processes, and support reproductive function and embryologic development.

The future looks bright as cannabinoid research, in the post-cannabis prohibition era, is finally able to provide additional discoveries regarding the role the endocannabinoid system plays in the pathogenesis of disease, and the maintenance of health [88].

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
