# Peer review of "The Endocannabinoid System of Animals"

_animals, 2019, doi:10.3390/ani9090686_

Round 1

Reviewer 1 Report

In this manuscript, the author provides a comprehensive overview of the Endocannabinoid system of animals and underlies the potential contribution to emerging therapeutics. The manuscript is clearly written and provides an explanation of studies in mammal species and invertebrate species, moreover, the author describes all the latest discoveries from clinical studies.

The novelty of the topic is that compared to previous reviews this manuscript gives an overall view describing the endocannabinoid system not only in mammals but also in invertebrates. However, the manuscript needs a minor revision in order to strengthen the arguments.

General comments:

Line 47-48 the author cites De Petrocellis but is missing the date, please include it. Line 58 the author needs to include a reference in this citation or rephrase it. Line 64 the author lists several biological functions where CB2 receptors participate but he includes only one reference, please add more references corresponding to the list made. Line 86 Paragraph 2 is too long can the author include a subparagraph as he did in paragraph 3. Line 94 when the author talks about “ECB production” does he refer to local production. Can the author clarify this point? Line 113 “the anatomical location of the CB1 receptors is found in dogs.” The author has to include a reference. Line114 “…studies have determined the number of CB1 receptors in hindbrain structures in the dog to far exceed those found in the human-animal”. The author has to include a reference. Line 117 the author is talking “studies” please include more references. Line 150 “Differences in the protein sequences of the CB2 receptor have been identified in the human, rat and canine receptors”. What about non-human primate? Is there any evidence? Line 178 “…embryo..” is this referred to the human embryo? The author should specify. Line 182 Can the author make a table describing the anatomical and cellular localization of CB1 receptors may be including also CB2 receptors localization. It would be easier for the reader to follow this. Paragraph 3.1.2 the author talks about CB2 receptors in dogs and humans what about other mammalian species? Can the author include other species? Line 221 the author is citing McPartland but the date is missing please include it. Line 235 the word hypothesizes should be “hypothesized”. Please correct it. Line 243 the word “This” should not be capital. Please correct it. Line 248 in the title there is a reference? Please correct this. Line 453 the author is talking about “two studies”, please include the references of the two studies. Line459 in the references list is missing #79. Please correct it.

Author Response

Line 47-48: Date added to Petrocellis reference.

Line 58-62: Reference added for this discovery, reference number updated to include this and additional references requested by this reviewer, and references added to list of references at end of paper. Sentence added to increase clarity of paragraph.

Line 64: References for each biological function added per reviewer request, citation numbering updated and references added to references at end of article.

Line 86: Expanded explanation in paragraph pointed out by reviewer and separated into sub-paragraphs, and added another  reference and updated reference numbering.

Line 94: See line 106 in revised version for explanation of local event of eCB production

Line 113 and line 114-same reference: Reference added for substantiation of difference in enumeration of eCB receptors in dog versus human. Now line 130-132.

Line 117: Reference added for US govt studies on dogs; now line 134.

Line 150: I could not find any studies of DNA sequences for non-human primates. Now line 177.

Line 178: Embryo species inserted with reference

Line 182: Table inserted per reviewer's preference. Now line 218. A separate Table has been inserted to substitute for the bulleted list that describes the CB2 receptor anatomical locations. References inserted for both tables. Now line 250.

Paragraph 3.1.2: To describe all of the other species that have had CB2 receptors localized is beyond the scope of this article that focuses on dogs.

Line 221: McPartland date inserted in references checked. Now lines 266 and 280.

Line 235: corrected spelling of hypothesis.

Line 235: corrected capitalization error. Now line 288

Line 248: Removed reference from title and inserted where appropriate

Line 453: Corrected this paragraph, removed one study that wasn't as definitive and kept and referenced the definitive study. Now line 515

Line 459: Reference corrected.

Reference number updated due to addition of another 11 references.

Reviewer 2 Report

It is an interesting review of the endocanabinoid system throughout the animal kingdom. It is very well written in very fluent English. I think it should be published in your prestigious magazine.

There are only three errors in the text:

line 114: compared to ..... of of (remove of) CB1

line 164: found to .... was found in The (the) fibers ... globus (cursive)

line 165: pallidus (cursive) and substantia nigra (cursive)

Author Response

line 114: deleted redundant word

line 163-164: lower case the; cursive of globus pallidus and substantia nigra, deleted redundant statement: possessed dense....

I do not understand why I would be italicizing these two anatomic structures of the brain and not the other anatomic structure listed earlier.

This manuscript is a resubmission of an earlier submission. The following is a list of the peer review reports and author responses from that submission.